# FRIDA-Clim v1.0.0: a Simple Climate Model with Process-Based Carbon Cycle used in the FRIDAv2.1 IAM

Christopher D. Wells<sup>1</sup>, Lennart Ramme<sup>2</sup>, Chris Smith<sup>3,4</sup>, Jannes Breier<sup>5,6,7</sup>, Adakudlu Muralidhar<sup>8</sup>, Chao Li<sup>2</sup>, Ada Gjermundsen<sup>8,9</sup>, William Alexander Schoenberg<sup>10,11</sup>, Benjamin Blanz<sup>12</sup>, Cecilie Mauritzen<sup>8</sup>

- <sup>1</sup>School of Earth and Environment, University of Leeds, Leeds, LS2 9JT, United Kingdom
  - <sup>2</sup>Max-Planck-Institute for Meteorology, Hamburg, Germany
  - <sup>3</sup>Department of Water and Climate, Vrije Universiteit Brussel, 1050 Brussels, Belgium
  - <sup>4</sup>Energy, Climate and Environment Program, International Institute for Applied Systems Analysis (IIASA), Laxenburg, Austria
- 10 <sup>5</sup>Potsdam Institute for Climate Impact Research, Potsdam, Germany
  - <sup>6</sup>Max Planck Institute of Geoanthropology, Jena, Germany
  - <sup>7</sup>Humboldt Universität zu Berlin, Department of Geography, Berlin, Germany
  - <sup>8</sup>Norwegian Meteorological Institute, Oslo, Norway
  - <sup>9</sup>University of Oslo, Oslo, Norway
- 15 <sup>10</sup>System Dynamics Group, University of Bergen, P.O. Box 7802, 5020 Bergen, Norway
  - <sup>11</sup>isee systems inc., 24 Hanover St. Suite 8A, Lebanon, New Hampshire 03766, USA
  - <sup>12</sup>Research Unit Sustainability and Climate Risks, University of Hamburg, Hamburg, Germany

Correspondence to: Christopher D. Wells (c.d.wells@leeds.ac.uk)

Abstract. The new global Feedback-based knowledge Repository for Integrated Assessments version 2.1 (FRIDAv2.1) Integrated Assessment Model (IAM) seeks to study the dynamics of the coupled human-Earth system. Connecting anthropogenic emissions to the resultant climate response is one part of this two-way feedback. This paper documents the Climate Module within FRIDAv2.1, of which a modified version is separately simulated as a standalone simple climate model termed FRIDA-Clim version 1.0. This approach, based loosely on the existing FaIR simple climate model, simulates the key radiative forcings and the resultant temperature response, with process-based representations of the carbon cycle across the ocean, land, and atmosphere. When connected within the FRIDA IAM, it features deep connections to the other modules, being affected by processes such as water use for irrigation and land use change. In both uses, coupled and uncoupled, its climate drivers are simplified as compared to FaIR, to allow for this reduced set of key drivers to be interactively simulated within FRIDA, tightly coupling the evolution of the social and climate systems within the full model. Both the Climate Module and FRIDA-Clim are fully calibrated to accurately reproduce observations of key climate variables, with a systematic exploration of the uncertainty in the climate response. Together with the rest of the FRIDA model, this module is used to incorporate climate change systematically in the FRIDA System Dynamics IAM. As a standalone climate model, FRIDA-Clim comprises a simple climate model, enabling fast calculation of the global climate response to forcing; to explore this, the response of the model to both idealised CO<sub>2</sub> emissions experiments and plausible

future scenarios is also presented here. This setup will allow FRIDA-Clim to contribute to inter-model simple climate modelling initiatives, helping to explore the structural uncertainty in this modelling domain.

#### 1 Introduction

Accurate, process-based modelling of the response of the climate to historical and future drivers is essential to study future climate policy and scenarios (Smith et al., 2024b). This modelling is undertaken across a range of levels of complexity, from detailed Earth System Models (ESMs) to lightweight simple climate models (SCMs). These groups of models are compared and studied in the Coupled Model Intercomparison Project (CMIP; Eyring et al., 2016) and the Reduced Complexity Model Intercomparison Project (RCMIP; Nicholls et al., 2020) respectively.

These models can be driven with exogenous emissions and other forcers, studying how the climate evolves independently of the social system responsible for them. The evolution of that social system, particularly the climate-impacting components, can be separately explored in an integrated assessment model (IAM), providing the inputs to drive climate models in CMIP and RCMIP frameworks.

Increasingly, in order to study the interactions between the climate and social systems, couplings between full complexity

ESMs and IAMs are being developed (Hao et al., 2023; Yokohata et al., 2020). Since ESMs are expensive to run, and
generally provide information with unnecessarily high levels of detail for IAMs, an SCM may be connected to an IAM to
provide fast simulations of the coupled system. Cost-benefit IAMs such as DICE include a climate module as part of the
optimization framework (Nordhaus, 2018), but are limited in their sectoral coverage, emissions species modelled, and
representation of the economy (Beckage et al., 2022; Donges et al., 2021; Wilson et al., 2021). The new Feedback-based
knowledge Repository for IntegrateD Assessments version 2.1 (FRIDAv2.1) model represents one such coupling of IAM
and SCM concepts, with its climate and social system models drawing on, but representing substantial modifications of,
existing IAMs and SCMs (Schoenberg et al., 2025a). Crucially, as a process-based IAM (Weyant, 2017) that includes
sectoral detail, FRIDAv2.1 incorporates a higher level of process complexity than cost-benefit IAMs.

FRIDAv2.1 is a global IAM, developed using system dynamics principles (Forrester, 1961; Sterman, 2000), with a focus on representing feedbacks between components of the coupled human-Earth system, at the expense of detail complexity (Schoenberg et al., 2025a). As such FRIDAv2.1 is conceptualised as a CHANS (coupled human and natural systems, e.g. Kramer et al., (2017)) model. The coupling between climate and society, with climate drivers from the social system affecting climate, and climate impacts consequently affecting the social systems, are therefore key components of the model. In this sense, while components of the framework were drawn from different models, they are conceptually similar within FRIDAv2.1. Each component of the model is designed to reflect a similar level of process detail, and to prioritise the

interactions between them. Both FRIDA-Clim and FRIDAv2.1 are publicly available, including the calibration procedure detailed here (see the Code Availability section).

This paper documents the Climate Module of FRIDAv2.1, which has at its core components from the FaIR SCM (Leach et al., 2021), but features an entirely new ocean and land carbon cycle implementation, a simplification of the number and representation of climate drivers, a representation of the dynamics of sea level rise (SLR), and a full recalibration. These changes were undertaken in order to better reflect the process-based system dynamics modelling philosophy of the FRIDA model, as well as the desire for a tractable process-based carbon cycle, as detailed in Section 2. The Climate Module has also been decoupled from the human system representation, calculating the climate response to exogenous forcings, in order to provide a new SCM named FRIDA-Clim, i.e. a model oriented towards initiatives such as a forthcoming phase of RCMIP. Both FRIDA-Clim and the integrated Climate Module are documented here. FRIDA-Clim aims to represent all global components of the carbon cycle, the radiative energy balance, and the relevant components of the water cycle, including, when coupled as the Climate Module within FRIDAv2.1, their interactions with the social system. As shorthand within this paper, the term "FRIDA" is often used when discussing concepts common to both models.

There are already many SCMs with a wide range in scope and complexity (Romero-Prieto et al., 2025), as explored in intermodel comparisons (Nicholls et al., 2021, 2020). FRIDA-Clim builds on one such widely-used SCM, the FaIR model, significantly adapting it by adding a coupled carbon cycle, as already incorporated in other SCMs (Romero-Prieto et al., 2025). FRIDA-Clim's uniqueness in the SCM field stems from its originating via a system dynamics modelling standpoint, and specifically its development as an integrated climate module within a fully-coupled IAM, lending it a focus on simplicity of inputs and connectability to the human system.

Section 2 details the representation of climate within the framework, including the coupling of FRIDA-Clim within FRIDAv2.1 as its Climate Module; Section 3 discusses how both versions are calibrated to historical climate observations, including the additional calibration steps when integrating into FRIDAv2.1. Section 4 explores the climate response of FRIDA-Clim under idealised CO<sub>2</sub> emissions experiments and future scenarios; Section 5 discusses the choices in the model creation process, and the relationship between FRIDA-Clim and the Climate Module; and Section 6 provides concluding remarks.

# 5 2 Model Description

This section details the representation of the climate system in FRIDA-Clim, and the differences when incorporated as the Climate Module in FRIDAv2.1, with the structure of each model and their relationships detailed in Figure 1. This representation begins with the imposition of anthropogenic emissions and other climate forcers, externally provided to



FRIDA-Clim but endogenously simulated within FRIDAv2.1 (Section 2.1), through their calculated effect on radiative forcing (Section 2.2) and consequent impact on global mean surface temperature anomalies (Section 2.3), modelled identically in FRIDA-Clim and the Climate Module. The process-based carbon cycle, essential to represent the flow of carbon through the Earth system, is then presented here for the land (Section 2.4) and ocean (Section 2.5); while the ocean component is the same between both models, the complex human-climate linkages within the land use sector necessitate a different approach between the standalone and integrated versions. The SLR module, while essential to the representation of climate damages, is presented elsewhere as a standalone SLR model (Ramme et al., 2025) and has no internal feedback to the climate system, and is therefore only briefly outlined here in Section 2.6.





Figure 1: Outline of the key components of the modelling framework, and the scope of each model. Components in grey boxes are calculated internally throughout the modelling framework, while striped boxes show quantities which are always externally provided. Hatched boxes are calculated endogenously in the fully-coupled FRIDAv2.1 IAM, but must be externally forced in the standalone FRIDA-Clim. Connections imply a causal link of the lefthand component on the righthand one, except the connection between the EBM and the rest of the model. Anthro. = anthropogenic.

Each step of the process after emissions are produced is subject to uncertainty analysis as part of the model calibration. The calibration process is detailed in Section 3, with parameters associated with Sections 2.2 - 2.5 listed with their ranges and described in Table S1.

While the FaIR SCM (Leach et al., 2021) has been coupled within an IAM in previous work (Smith et al., 2023), in FRIDAv2.1 it is substantially altered and fully subsumed within the model via the Climate Module. The radiative forcing and temperature response is loosely based on FaIR, with the key differences being the reduced number of forcing species, and the recalibration. The land carbon cycle in part reflects a simplified representation of the LPJmL land model (Schaphoff et al., 2018). The ocean carbon and chemistry representation is based on the carbon cycle model of Lenton, (2000) and the LOSCAR model (Li et al., 2024; Zeebe, 2012).

The philosophy of FRIDA-Clim, in keeping with that of FRIDAv2.1 as a whole, is that only the minimum level of detail in the climate system required in order to adequately reproduce historical and future expected global climate dynamics should be represented. This "minimal required climate" should therefore only feature the most important emissions species and forcers. Relatedly, in order to maintain scenario coherence as well as study the feedback connections between the climate drivers, the climate response, and the social system when coupled within FRIDAv2.1, the dynamics of these forcers should be endogenously simulated within the IAM; only in exceptional cases should exogenous timeseries be utilised. As described in Section 2.2, in FRIDAv2.1, only Solar, Volcanic, and Montreal Gases forcings, as well as Montreal Gases' effect on Ozone, are implemented exogenously as inputs to the Climate Module, due to their assumed independence from the social system. All other climate drivers are explicitly modelled within FRIDAv2.1, allowing for their interactive response to changes in the components of the social system which drive them. When run as the standalone FRIDA-Clim model, all forcers, including emissions and land use dynamics, must be exogenously imposed. FRIDA-Clim is designed to be initialised in pre-industrial times (1750 here), while simulations in FRIDAv2.1 begin in 1980.

# 2.1 Emissions and other Climate Drivers

Figure 2 shows the flow of information within the climate system as modelled here. "Anthropogenic Climate Driver(s)", shown in black, are exogenously imposed in FRIDA-Clim, but interactively simulated within FRIDAv2.1, with each module directly generating one or more climate driver(s) (except Demographics and Behavioural Change which have indirect effects






via other modules); multiple sources of emissions are collated into total overall emissions, for further use within the Climate Module. These seven Anthropogenic Climate Driver(s) are listed in Table 1, along with their contributing modules and sources of calibration data in FRIDAv2.1. These comprise emissions of five conventional species - CO<sub>2</sub>, CH<sub>4</sub>, SO<sub>2</sub>, N<sub>2</sub>O, and HFCs (treated as HFC134a-equivalent) - as well as H<sub>2</sub>O emissions from irrigation, and the change in the surface albedo due to shifts in land use.

These seven drivers are then used to derive nine overall interactively simulated anthropogenic radiative forcings, with an exogenous contribution from Montreal Protocol-controlled gases' effect on stratospheric ozone depletion, quantified as the equivalent effective stratospheric chlorine (EESC) concentration. Three exogenous radiative forcings are added to this set the direct forcing from Montreal protocol-controlled gases, and natural volcanic and solar forcings. The 12 total effective radiative forcing (ERF) categories are each described in the next section.

Four variables shown in Figure 2 relate to species whose emissions are not modelled in a process oriented fashion, and are instead modelled using other emissions species as drivers in FRIDA: emissions of NOx, VOCs, and CO, and the forcing due to black carbon (BC) deposition on snow (see Section 2.2). These four variables were deemed relevant enough for the climate - the emissions due to their effect on ozone forcing and CH<sub>4</sub> lifetime, and the BC deposition on snow forcing as a relevant climate radiative effect (Forster et al., 2021; Szopa et al., 2021) - to be incorporated within the model, but the simulation of their process-based drivers was not possible at the level of sectoral detail currently modelled in FRIDA. Because of this, their evolution is instead approximated using simple regression relationships, using some of the actual simulated direct outputs from the IPCC Sixth Assessment Report database of IAM-derived emissions scenarios (Byers et al., 2022) (Table 1) as their predictors. Using historical data, and checked for plausibility using the future scenarios database, several candidate regression models were considered for each variable, with a linear relationship selected (see Figure S1 for the fits). NO<sub>x</sub> emissions were split into two components, with their AFOLU emissions predicted by SO<sub>2</sub> emissions, and their larger non-AFOLU component from N<sub>2</sub>O non-AFOLU emissions. Emissions of VOCs and CO are both linearly modelled with CH<sub>4</sub> emissions, and the BC deposition on snow forcing is predicted by both SO<sub>2</sub> and CO<sub>2</sub> AFOLU emissions. While primarily informed by the data correlations, these connections were deemed roughly plausible on the process-level, as similar processes connect the predictors with the variables they are used to drive. This idea follows the "infilling" logic used to generate a full set of relevant climate forcers from a limited set of inputs, as is often used in the climate assessment of IAM-derived emissions scenarios (IPCC, 2022; Kikstra et al., 2022; Lamboll et al., 2020; Smith et al., 2023).



Figure 2: Overview of the calculation of radiative forcings within this modelling framework; see Sections 2.1-2.2 for details. Anthropogenic climate drivers (black) are interactively simulated within FRIDAv2.1 in the non-Climate modules as shown on the left and fed into the Climate Module; these drivers are externally provided to FRIDA-Clim when run as a standalone model. Similarly, climate impacts are only calculated in FRIDAv2.1, and then connected back to the rest of the IAM, completing the feedback loop between the components of the human-Earth system. The grey lines comprise the set of interactions modelled in FRIDA-Clim. Atmospheric variables calculated within the climate model are shown in grey boxes, with internal climate feedbacks shown with green hatched boxes. Striped grey boxes denote exogenously imposed variables. The total radiative forcing feeds into the energy balance model. Information flows from left to right; the coloured and grey lines denote an influence of the left connection onto the right one. Dashed lines indicate that a subset of the left connection is used as the influencer, with both cases relating to the regression-based predictions (see Section 2.1).

| Anthropogenic<br>Climate Driver | Associated Radiative Forcing(s) | Source<br>Module | Full-Model Calibration Data                                                                                             |
|---------------------------------|---------------------------------|------------------|-------------------------------------------------------------------------------------------------------------------------|
| HFC Emissions                   | HFC                             | Energy           | HFC134a-equivalent emissions estimated from concentrations based on Indicators of Global Climate Change (IGCC; Smith et |

|                             |                                                      |                           | al., 2024a) with radiative properties from (Hodnebrog et al., 2020) (see Section 2.2.7).                                                                                         |
|-----------------------------|------------------------------------------------------|---------------------------|----------------------------------------------------------------------------------------------------------------------------------------------------------------------------------|
| CO <sub>2</sub> Emissions   | CO <sub>2</sub><br>BC Snow                           | Energy                    | Global Carbon Project, v2024, v1.0 (Friedlingstein et al., 2025) - Fossil excluding carbonation, minus cement.                                                                   |
|                             |                                                      | Resources                 | Global Carbon Project, v2024, v1.0 (Friedlingstein et al., 2025) - cement                                                                                                        |
|                             |                                                      | Land Use &<br>Agriculture | Global Carbon Project, v2024, v1.0 (Friedlingstein et al., 2025) - land-use change                                                                                               |
| SO <sub>2</sub> Emissions   | Aerosol<br>BC Snow<br>Ozone                          | Energy                    | Community Emissions Data System (CEDS) v2024_07_08 (Hoesly et al., 2024) - all categories.                                                                                       |
|                             |                                                      | Land Use & Agriculture    | GFED (van Marle et al., 2017) historical average, i.e. constant.                                                                                                                 |
| N <sub>2</sub> O Emissions* | N <sub>2</sub> O<br>Ozone                            | Energy                    | PRIMAP-hist v2.6 "no_rounding" (Gütschow et al., 2024) - categories 1 (Energy) and 2 (Industrial Processes and Product Use).                                                     |
|                             |                                                      | Economy                   | PRIMAP-hist v2.6 "no_rounding" (Gütschow et al., 2024) - categories 4 (Waste) and 5 (Other).                                                                                     |
|                             |                                                      | Land Use & Agriculture    | PRIMAP-hist v2.6 "no_rounding" (Gütschow et al., 2024) - category 3 (Agriculture, Forestry and Other Land Use) plus GFED BB4CMIP (van Marle et al., 2017) historically-averaged. |
| CH <sub>4</sub> Emissions   | CH <sub>4</sub> Ozone Stratospheric H <sub>2</sub> O | Energy                    | CEDS v2024_07_08 (Hoesly et al., 2024) - categories 1, 2, 7A.                                                                                                                    |
|                             |                                                      | Economy                   | CEDS v2024_07_08 (Hoesly et al., 2024) - categories 5, 6, 7BC.                                                                                                                   |
|                             |                                                      | Land Use & Agriculture    | CEDS v2024_07_08 (Hoesly et al., 2024) - category 3, plus GFED (van Marle et al., 2017) historical average.                                                                      |

| Irrigation Emissions | H <sub>2</sub> O | Land Use | Land Use &<br>Agriculture | (Forster et al., 2025) |
|----------------------|------------------|----------|---------------------------|------------------------|
| Land Albedo          |                  | Land Use | Land Use &<br>Agriculture | (Ghimire et al., 2014) |

Table 1. Anthropogenic climate drivers output from non-climate modules in FRIDA which feed into the Climate Module, along with their downstream radiative forcings, source module(s), and datasets used in the calibration of the full model where relevant (see Section 3.2). Note that in the standalone FRIDA-Clim model and the calibration of the Climate Module, these drivers are exogenously imposed. See the Code Availability section for the data processing details. \*N<sub>2</sub>O emissions were increased by 7% to better match observations using IPCC AR6 best estimate atmospheric lifetime, as done in FaIR (Smith et al., 2024b).

## 190 2.2 Effective Radiative Forcings

The simulation of Effective Radiative Forcings (ERFs) is performed identically in both FRIDA-Clim and the FRIDAv2.1 Climate Module, with their calculations described here, and equations given for some key forcings. See Table S2 for the units and values of specific parameters where not provided in-text.

### 2.2.1 CO<sub>2</sub>

The concentration of atmospheric  $CO_2$  is calculated as part of the full carbon cycle within the climate representation. Three fluxes of  $CO_2$  flow into or out of the atmosphere: anthropogenic emissions (Figure 2 and Table 1), land carbon flux (Section 2.4), and the air-sea  $CO_2$  flux (Section 2.5). The resultant  $CO_2$  concentration  $CO_2(t)$  is converted to a forcing  $(ERF_{CO_2})$  using the combined logarithmic and square-root parameterisation of Leach et al., (2021), with a 5% enhancement to account for the fast tropospheric feedbacks associated with the calculation of ERF (Forster et al., 2016):

$$ERF_{CO_2} = \left( f_1^{CO_2} \ln \left( \frac{cO_2(t)}{cO_2(1750)} \right) + f_3^{CO_2} \left( \sqrt{CO_2(t)} - \sqrt{CO_2(1750)} \right) \right) * 1.05,$$
(1)

with  $CO_2(1750)$  the concentration in 1750, and  $f_1^{CO_2}$ ,  $f_3^{CO_2}$  the logarithmic and square-root parameters respectively.

# 2.2.2 CH<sub>4</sub>

Due to the complex and heterogenous chemical processes, CH<sub>4</sub>, unlike CO<sub>2</sub>, is not tracked through the Earth system in FRIDA; instead, FRIDA-Clim and FRIDA's climate module retain the one-box atmospheric decay of FaIR (Smith et al., 2024b). In this representation, emissions of CH<sub>4</sub> accumulate in the atmosphere, and decay with a variable lifetime. As a chemically active species, CH<sub>4</sub>'s lifetime is affected by several species: concentrations of N<sub>2</sub>O and EESC, and emissions of VOC and NO<sub>x</sub> - increases of VOC emissions act to increase CH<sub>4</sub>'s lifetime, with the other species instead decreasing it. In

addition, increases in temperature accelerate the chemical reactions which break down CH<sub>4</sub>, decreasing its lifetime. With the CH<sub>4</sub> concentration calculated, its forcing  $ERF_{CH_4}$  is computed via the square-root parameterisation of Leach et al., (2021) used in FaIR, with a 14% reduction accounting for the fast tropospheric response:

$$ERF_{CH_4} = f_3^{CH_4} (\sqrt{CH_4(t)} - \sqrt{CH_4(1750)}) * 0.86,$$
 (2)

# 2.2.3 N<sub>2</sub>O

The representation of  $N_2O$  is similar to that of CH<sub>4</sub>. The key difference is that, as a long-lived, non-chemically active species, the lifetime of  $N_2O$  is only affected by its cumulative emissions, with a decreasing effect on the lifetime. As for CH<sub>4</sub>, its ERF tracks the square root of the concentration following Leach et al., (2021), with the parameter  $f_3^{N_2O}$  and a 7% increase to incorporate the tropospheric feedbacks.

#### 2.2.4 Aerosols

The ERF components from both aerosol-radiation and aerosol-cloud interactions are calculated in FRIDA. They are both functions of  $SO_2$  emissions  $ERF_{SO_2}$ , with a linear relationship for the aerosol-radiation interactions and a logarithmic relationship for the aerosol-cloud interactions:

$$ERF_{SO_2} = f_1^{ACI} \ln(1 + F_{SO_2}E(t)) + f_1^{ARI}E(t), \qquad (3)$$

With  $f_1^{ACI}$  and  $f_1^{ARI}$  the cloud and radiation scaling parameters,  $F_{SO_2}$  the cloud logarithmic parameter, and the corresponding pre-industrial baseline values separately subtracted.

It should be noted that, although SO<sub>2</sub> is the only aerosol species explicitly modelled in FRIDA, the calibration of the aerosol forcing to present-day estimates (see Section 3) causes SO<sub>2</sub> to represent a proxy for all contributors to these forcings, similarly to Stevens, (2015). In this way, the present-day global ERF impact of aerosols is ensured to correspond to the IPCC best estimate, even though only a single species is included explicitly.

# 2.2.5 Ozone



Ozone is not directly emitted, but is radiatively active and produced via reactions of anthropogenically altered species, therefore contributing a radiative forcing via human influence (Thornhill et al., 2021). In FRIDA, its forcing is contributed to by concentrations of CH<sub>4</sub>, N<sub>2</sub>O, and Montreal Protocol-controlled gases, with the latter effect negative; and by emissions of VOC, CO, and NO<sub>x</sub>, with the magnitude of these effects being likely positive but encompassing zero. Montreal Gases deplete the ozone layer in the stratosphere (as does N<sub>2</sub>O to a small extent), and other species promote the formation of ozone






in the troposphere. Similarly to CH<sub>4</sub>, increased temperatures act to reduce Ozone's ERF through the increased formation of the OH radical, a tropospheric ozone sink, through water vapour dissociation (warmer atmosphere holds more water vapour through the Clausius-Clapeyron relationship (Suzuki et al., 2017)). The effect of Montreal Gases on Ozone forcing is via its effect on EESC, which is exogenously imposed, calculated using the FaIR model to estimate the effect of historical and future assumed concentrations of the relevant Montreal Protocol-controlled gases. This assumes, as in many conventional IAM scenarios, that the Montreal Protocol continues to be effective in strictly limiting emissions of ozone-depleting gases leading to a steady recovery of the stratospheric ozone layer. The historical concentrations of Montreal Gases are taken from the Indicators of Global Climate Change 2023 (Smith et al., 2024a), with the future data from the medium emissions scenario ssp245 (note the SSPs give identical concentrations for most Montreal Gas species concentrations), scaled to match the updated historical observations in 2022.

# **2.2.6** Land Use

The coupling of the climate to the human system in FRIDA allows for a more process-based representation of land usedriven forcings than FaIR, which represents the forcing due to land use change as a linear function of cumulative land-based CO<sub>2</sub> emissions. Two separate land use forcings are calculated in FRIDA.

Firstly, the change in albedo due to changes in land use is calculated in FRIDA, with the albedo shift calculated from the land use stocks. The land albedo is calculated as the area-weighted sum of the albedos for each land type - cropland (0.22), grassland (0.21), mature forest (0.15), and young forest (0.18; average of grassland and mature forest). Converting to the global surface albedo is not necessary, since the forcing response to albedo change is calibrated to give the present-day best-estimate response. The associated ERF is modelled as a linear function of the change in this albedo, with the sensitivity in the prior ensemble (see Section 3) chosen to give the present-day value and distribution from Forster et al., (2021) of -0.15  $\pm$  0.1 W m<sup>-2</sup>. Historically, an overall shift from forests to grasslands and croplands via deforestation has increased the albedo, generating a negative present-day forcing.

Secondly, the emission of H<sub>2</sub>O from water used for irrigation is simulated in FRIDA. This water can either act as a positive-forcing greenhouse gas, in its vapour state, or generate a negative forcing due to an increase in low cloud upon condensation (Gormley-Gallagher et al., 2022; Sherwood et al., 2018). The net effect of this is uncertain, with the central estimate negative, suggesting a larger role for the increase in low clouds (Sherwood et al., 2018). The overall effect is modelled with a single parameter in FRIDA, a linear function of the irrigation emissions, which in turn is scaled from agricultural water withdrawal. The distribution of prior parameters is selected to reproduce the estimate of Forster et al., (2021), following Sherwood et al., (2018).

The 1980 values of both land use forcings are calculated, and these are input to FRIDA as constant parameters.

# 2.2.7 HFCs



One substantial simplification in FRIDA, in comparison to FaIR, is the aggregation of the Kyoto Protocol gases, which in FRIDA and FRIDA-Clim are expressed as an equivalent level of a single, representative species HFC134a, or HFC134a-eq. This was necessary due to the philosophy of interactively simulating at the process-level the emissions of each relevant species, which would not be possible for the many minor GHGs represented in FaIR. Driven by cooling energy demand (Section 2.1), HFC134a-eq is modelled with a single atmospheric box, like CH<sub>4</sub> and N<sub>2</sub>O, but with a fixed lifetime, and a linear relation to its ERF.

Historical HFC134a-eq emissions for model calibration were estimated based on the annual emissions required to reproduce HFC134a-eq concentrations, themselves calculated by weighting the concentrations of the contributing species by their radiative efficiencies, from (Hodnebrog et al., 2020).

# 2.2.8 Montreal Protocol-controlled gases

The remainder of the minor GHGs in FaIR comprise the Montreal Protocol-controlled gases, refrigerants which are subject to strict controls under the Montreal protocol (Egorova et al., 2023). While it is possible that compliance issues may drive multiple plausible future levels of these gases, the modelling of these dynamics is not within the scope of FRIDA. Therefore, a single, exogenous, future decline in the ERF from the Montreal Gases is imposed in FRIDA, calculated using the FaIR model as for EESC (Section 2.2.5). Together with their effect on Ozone forcing (2.2.5), and the HFC134a-eq forcing (2.2.7), this comprises the effect of minor GHGs in FRIDA.

# 2.2.9 Stratospheric H2O from CH4 oxidation

One major loss pathway of CH<sub>4</sub> is oxidation to H<sub>2</sub>O in the stratosphere, which then imposes a slight positive radiative effect.

This is accounted for in FRIDA by driving a small stratospheric water vapour forcing, linear in the CH<sub>4</sub> forcing.

# 2.2.10 Black carbon deposition on snow

Depositions of (dark) black carbon (BC) aerosol on (light) snow can result in an additional slight positive ERF, due to the localised absorption of radiation. Since BC emissions are not simulated in FRIDA (and noting that global emissions may not be an accurate predictor of this regional effect in any case), this effect is instead captured using the linear regression approach described in Section 2.1. The optimal predictors in this case were found to be CO<sub>2</sub> AFOLU emissions and Total SO<sub>2</sub> emissions (see Figure S1).

# 2.2.11 Volcanic



The first exogenous natural forcing represented in FRIDA is that due to volcanic eruptions. The process follows that of FaIR - and used in IPCC AR6 (Smith et al., 2021) - in which the historical volcanic forcing is utilised until present-day, with a constant value applied in future. Since the forcing should be compared to the pre-industrial long-term equilibrium climate, which contained a level of volcanic forcing assumed constant in time (when averaged over decades), the forcing timeseries is offset so as to have zero overall forcing over the historical timeperiod. Thus, the historical timeseries of volcanic forcing manifests as a small positive ERF in most years, punctuated by the negative eruption-based forcings which this positive value is designed to offset. The future value is then set to zero (linearly reduced from the present value over a decade), due to the lack of knowledge about future volcanic eruptions. To account for the lower contribution to surface temperature change than implied by its ERF, this forcing is adjusted by multiplying by an efficacy (Hansen et al., 2005) of 0.6.

#### 2.2.12 Solar

The second exogenous natural forcing in FRIDA, and final forcing overall, is the variation in solar activity. Unlike for volcanic forcings, this variation is somewhat predictable on a roughly 11-year cycle, and so is imposed as a time-varying ERF throughout the whole simulation period, using the CMIP6 solar forcing timeseries for the historical and future period (Nicholls and Lewis, 2021). To explore the uncertainty in the solar forcing, scaling factors modifying the amplitude and long-term trend - centred around 1 and 0 respectively - are applied to generate the overall Solar ERF.

# 2.3 Energy Balance Model

Once the total radiative forcing is calculated, as the sum of the 12 forcings detailed above, this energy imbalance is input to a three-layer energy balance model (EBM) representation of the Earth system, following the approach in FaIR (Cummins et al., 2020). This representation is identical between FRIDA-Clim and FRIDAv2.1's Climate Module. In this formulation, the Earth system is split into three vertically-resolved layers with varying heat capacities and rates of energy exchange, the first of which experiences the incident *ERF*, with the energy imbalance exchanged between connecting layers:

$$C_1 \frac{dT_1}{dt} = ERF - \kappa_1 T_1 - \kappa_2 (T_1 - T_2), \tag{4}$$

$$C_2 \frac{dT_2}{dt} = \kappa_2 (T_1 - T_2) - \varepsilon \kappa_3 (T_2 - T_3),$$
 (5)

$$C_3 \frac{dT_3}{dt} = \kappa_3 (T_2 - T_3), \tag{6}$$



with  $T_i$ ,  $C_i$ ,  $\kappa_i$  the temperature anomaly, heat capacities, and heat transfer coefficients of each layer, with units K, W m<sup>-2</sup> yr<sup>-1</sup> K<sup>-1</sup>, and W m<sup>-2</sup> K<sup>-1</sup> respectively. The parameter  $\varepsilon$  is the dimensionless deep ocean uptake efficacy factor, modifying the exchange with the deep ocean, thereby allowing for a representation of the varying transient response of the system. The first layer can concretely be associated with the Earth's surface and ocean mixed layer, as this response is calibrated to reproduce the historically observed GMST (see Section 3); the two lower layers conceptually represent the intermediate and deep oceans respectively, though the lack of calibration data renders the exact association ambiguous.

# 2.4 Land carbon cycle

The land carbon cycle represents the domain with the largest differences in approaches between FRIDA-Clim and the Climate Module, due to the complex coupling of human and climate processes within the land system; the representation in FRIDA-Clim is based on that in the LPJmL model (Schaphoff et al., 2018), with substantial modifications. FRIDAv2.1 calculates crop production (as meeting demand), driven by human and climate drivers, but in FRIDA-Clim the lack of modelled human drivers necessitates that this variable is externally supplied. Similarly, land use transitions and forest cutting rates are endogenous to FRIDAv2.1 but exogenous to FRIDA-Clim. Other processes, such as grass and forest growth and soil carbon deposition and decay, have the same representations, with only the baseline values varying between the models due to their different initialisation years.

FRIDA separates the land into three main land use types: cropland, grassland, and forests (with forests subdivided into young and mature forests), as well as a small stock of degraded land. The model represents the key flows of carbon between and amongst the land and the atmosphere, as shown in Figure 3. The anthropogenic drivers of land usage - i.e. land use change and crop production - are explicitly modelled within the Land Use and Agriculture module in FRIDAv2.1. In FRIDA-Clim, the effects of these drivers must be provided as external timeseries (Section 2.7).

Figure 3: Schematic representing the stocks and flows of the carbon cycle within FRIDAv2.1 and FRIDA-Clim. Anthropogenic emissions enter the atmosphere, which causes imbalances in the global equilibrium leading to flow of carbon across the air-sea interface, altering the two ocean surface boxes, which in turn interact with the two deeper layers (Section 2.5). On land, uptake of carbon varies by land type, with biomass stocks of two age-based forests, and produced matter transitioning to the soil stocks. Soil stocks in turn decay and are re-emitted, and switch land types when land use changes occur. Note that land transitions shift soil stocks for fast and slow stocks separately.

The response of the land system to climate change - via the effects of temperature and CO<sub>2</sub> on net primary production (NPP), soil carbon, and forest biomass - are simulated in the model, as part of the carbon cycle. In FRIDAv2.1, these dynamics are situated in the Land Use and Agriculture Module, but are described here as key internal climate feedbacks.

# 2.4.1 Net Primary Productivity


Carbon enters the land system by the uptake of plants via photosynthesis. Changes in this, expected under a changing climate (Knorr et al., 2005), are represented by determining the *NPP* per area (in GtC Mha<sup>-1</sup> yr<sup>-1</sup>) as a function of surface temperature and CO<sub>2</sub> concentration:

$$NPP = NPP_{Rase}(1 + a\Delta T + b\Delta T^2 + c\Delta CO_2), \qquad (7)$$

where Base represents the initialisation year of the model - 1750 in FRIDA-Clim, and 1980 in FRIDAv2.1 (and hence its Climate Module) - and  $\Delta T$  and  $\Delta CO_2$  represent changes in the global mean temperature (K) and CO<sub>2</sub> concentration (ppm)






compared to this baseline. In the FRIDAv2.1 Climate Module, NPP is interactively calculated in this way in all three main land use type (cropland, grassland, and forest), with land type-dependent baseline NPP values and sensitivity parameters a, b and c varied in the calibration ensemble. Cropland NPP is additionally a function of crop fertilizer use, irrigation, and soil carbon in the fully coupled IAM. However, in FRIDA-Clim, these cropland anthropogenic drivers are not simulated. Instead, cropland production (in GtC) must be directly provided; this is then used to calculate cropland NPP. In this way, in FRIDA-Clim, the climate only interactively affects grassland and forest NPP. Since literature-based estimates for the sensitivity parameters that are consistent with this framing of global land use were not available, they are varied as part of the internal calibration of FRIDAv2.1 (Section 3).

In both models, the resultant cropland NPP is split between usable crop production and crop residues. Part of the crop residues remain as litter and contribute to soil carbon, with the rest designated for other human purposes. The produced crops and non-litter residues are assumed to be consumed on a short timescale and directly translated into CO<sub>2</sub> emissions.

Grassland is partly used as pasture, which has a similar effect on the carbon cycle as crop production on cropland: the carbon going into animal grazing is directly released into the atmosphere, while the remaining part of grassland NPP enters the soil as litter. Therefore, plant growth on grassland takes place on an annual basis. In contrast, in forests part of the NPP contributes to the long-term accumulation of aboveground biomass (AGB), one of the largest biospheric carbon pools alongside soil carbon (Erb et al., 2018; Friedlingstein et al., 2025). Forests are further divided into young and mature forest, with any land that is transformed into forest entering the young forest stock, and young forest aging into mature forests. For simplicity, it is assumed that young forests use part of their NPP over a certain period to build up AGB, while mature forests are only maintained, such that all mature forest NPP contributes to litter and thus to soil carbon (Pregitzer and Euskirchen, 2004).

The global averaged aboveground biomass per forest area is affected by climate change, assuming a quadratic relationship to global temperatures:

$$S_{AGR} = d\Delta T + e\Delta T^2, \qquad (8)$$

with  $S_{AGB}$  the (dimensionless) scaling factor for a change in maximum aboveground biomass, and  $\Delta T$  the surface temperature anomaly (K). This idealised implementation and functional form were utilised because of complex competing effects of climate change on AGB: while vegetation can spread and grow larger under climate change (Berner and Goetz, 2022; Cortés et al., 2021; Xu et al., 2015), represented by the positive linear temperature contribution (d), processes such as desertification and increasing fires (e.g. Brown and Johnstone, 2011; Thornley and Cannell, 2004) under global warming are not represented explicitly in FRIDA yet, instead incorporated in an idealised manner here via a negative quadratic term (e). The parameters were calibrated internally within the model, together with those of forest and grassland net primary production,

targeting the evolution of the natural land carbon sink and terrestrial carbon balance from the Global Carbon Project 400 (Friedlingstein et al., 2025).

#### 2.4.2 Soil Carbon


Soil carbon is represented as two carbon stocks for each land use type - those with fast or slow decomposition rates, under a framework based on the LPJmL land use model (Schaphoff et al., 2018; particularly their Equation 45, with values and their uncertainties slightly re-calibrated to ensure the module more closely matches observations when coupled with the rest of the FRIDA-Clim structure; see Section 3). The plant litter left on the soil, which varies in each land use type due to different plant usages, decomposes modified by a function of global mean temperature:

$$S_D = e^{(e_0(\frac{1}{R} + 10) - \frac{1}{R + Ta})}, \tag{9}$$

where  $S_D$  is the scaling factor on the natural decay rate,  $T_a$  is the global mean absolute temperature in °C, and  $e_0$  and R are the function parameters in K. The soil carbon then decays at an annual rate D (yr<sup>-1</sup>):

$$D = 1 - e^{-\tau_i S_D}, (10)$$

with  $\tau_i$  the natural decay rate of soil carbon component *i* (fast, slow, or litter; units yr<sup>-1</sup>). These decay rates are independent of land-use type, and the respective decomposed carbon is emitted as  $CO_2$  into the atmosphere. For litter, the decay rate defines the fraction of carbon that decomposes before entering the soil, and is therefore not added to any soil carbon pool.

In isolation, this represents a separate carbon cycle for each land use type. However, land use transitions - interactively modelled in FRIDAv2.1, and externally imposed in FRIDA-Clim - cause the corresponding shares of land and soil carbon to also be transferred. In order to diagnose the effect of land use transitions on the carbon cycle, the committed carbon gain or loss for each transition is inferred assuming that the transitioned area will adapt to the average soil carbon density of the new land type. These committed carbon changes from all land-use transitions enter a global stock of total committed soil carbon loss due to land-use transitions. This stock is assumed to adapt (decay) on an adaptation timescale of ten years, and the corresponding emissions are assumed to be part of the human-driven "food and land-use" (FLU) emissions, as described in the following section. It should be noted that the actual soil decay happens in the corresponding pools of each land-use type, and that this accounting is a purely diagnostic way of separating human-driven from natural soil carbon cycle responses.

If a forest is cleared for cropland or grassland, the corresponding AGB carbon is released into the atmosphere, via the assumption that the material is burned or decomposed. Hence, there is no anthropogenic stock of wooden carbon, which under a more complex representation could account for carbon in wooden buildings or furniture, currently responsible for around 0.03-0.3GtC yr<sup>-1</sup> carbon removal from forests (Kayo et al., 2021). Similarly, the AGB released by cutting mature forest is emitted, while the respective areas and soil carbon pools are transferred from mature to young forest stocks. Lastly,




soil carbon is tracked for degraded land, including transition effects, but since degraded land NPP is very small (scaled as 1-10% of grassland NPP per area; see Table S1), these stocks mostly decay without much replenishing.

# 2.4.3 Terrestrial Carbon Balance

The terrestrial carbon balance (TCB) tracks the overall change of carbon on land, via the processes described above (plus a small constant annual peatland carbon sink of 0.3 GtC yr<sup>-1</sup> (Gallego-Sala et al., 2018; Loisel et al., 2021)).

For analysis and calibration purposes, the TCB is separated into a natural component ("land carbon sink"), responding to the changing climate, and a human component ("food and land-use (FLU) emissions"), driven by food production and land-use changes. FLU emissions are calculated by separating out the human-driven processes from the TCB, similar to other bookkeeping approaches (e.g. Hong et al., (2021)), and are defined here as the sum of the loss in aboveground biomass from forest clearing and cutting, the build-up of aboveground biomass and soil carbon in young forests, and the annually realized soil carbon changes driven by land use transitions. The natural land carbon sink is then simply the sum of both (the TCB is defined as positive downwards, while FLU emissions are positive upwards).

# 2.5 Ocean Carbon Cycle

The inclusion of a process-based land carbon cycle necessitates the inclusion of a separate ocean carbon cycle model as well, because the implementation in FaIR does not contain ocean carbon. In both FRIDA-Clim and the FRIDAv2.1 Climate Module, the ocean carbon cycle is modelled in a process-based manner using a four-box model of carbon in the ocean (Figure 3), based on the model proposed by Lenton (2000), with the ocean advection scheme and carbonate chemistry formulation based on those from the iLOSCAR model (Li et al., 2024; Zeebe, 2012).

The four boxes of ocean carbon represent the low-latitude (warm) and high-latitude (cold) surface ocean, the intermediate depth ocean and the deep ocean (Lenton, 2000). The warm surface ocean box represents 85% of the ocean's surface, with the cold surface ocean box covering the remaining 15%. The thickness of the two surface ocean boxes, as well as the thickness of the intermediate layer, are varied independently during the model calibration phase. The volume of the deep ocean box is calculated for each combination of level thicknesses of the above layers to reach an overall ocean volume of 1.36 billion km<sup>3</sup> (Lenton, 2000).

Within the ocean, carbon is redistributed by three distinct processes: mixing, overturning circulation, and biological processes (see Figure 3). Mixing occurs between the warm surface and the intermediate ocean layer and between the cold surface and the deep ocean, respectively. The overturning circulation transports water from the cold surface ocean into the deep ocean and via the intermediate ocean layer back into the cold surface ocean layer. This transport scheme is based on that in the iLOSCAR model, but the iLOSCAR model further separates the warm surface, intermediate and deep ocean into



Pacific, Atlantic and Indian Ocean sectors. The advection and mixing scheme of the iLOSCAR model was found to be more suitable for FRIDA than that from Lenton (2000), because it is slightly simpler and performed better at reproducing Earth system model output in an initial testing phase. The mixing and advection are applied via constant parameters that prescribe mass fluxes between the boxes. The mass fluxes are combined with the volume of the ocean boxes and their respective carbon content to produce fluxes of carbon between the model boxes, with mixing parameters and overturning strengths varied independently during the model calibration.

The representation of biological processes is implemented via a downward transport of carbon from the surface boxes into the intermediate layer ocean, representing the sinking of organic particles. A fraction of the carbon sinking down from the cold surface ocean is transferred directly into the deep ocean, while the downward transport from the warm surface ocean is assumed to be completely remineralized within the intermediate layer. All three parameters - warm and cold surface ocean carbon export, and the transfer efficiency - are varied independently during the model calibration phase, with the prescribed ranges informed by ESM output.

The two surface ocean boxes are in contact with the atmosphere and exchange carbon at a rate  $F_{air-sea}^i(GtC)$  based on the difference between their partial pressure of  $CO_2$  ( $pCO_{2,ocean}$ ) and the atmospheric  $CO_2$  concentration ( $pCO_{2,atm}$ ), modified by each box's area:

$$F_{air-sea}^{i} = kA_{oceans}F^{i}(pCO_{2.atm} - pCO_{2.ocean}^{i}).$$

$$\tag{11}$$

The index *i* represents the two surface ocean boxes, and *k* is the gas exchange coefficient, for which a value of 0.06 mol m<sup>-2</sup> yr<sup>-1</sup>ppm<sup>-1</sup> is used, taken from the iLOSCAR model (Li et al., 2024; Zeebe, 2012). A<sub>oceans</sub> is the total ocean area, with F<sup>i</sup> the area fraction of box *i*. The calculation of the atmospheric CO<sub>2</sub> concentration is a simple conversion of the atmospheric carbon content into ppm, but the conversion of the carbon content of the surface ocean boxes to a partial pressure of CO<sub>2</sub> requires a representation of carbonate chemistry in seawater. For this the version implemented in the iLOSCAR model is used, itself based on that from Follows et al., (2006). FRIDA builds on this previous literature with one difference. Typically, solving the carbonate chemistry system requires iteratively solving for the concentration of hydrogen ions in seawater, from which the ocean's pH can be deducted, which is required to calculate the separation of dissolved inorganic carbon into aqueous CO<sub>2</sub>, bicarbonate and carbonate. However, the high level of aggregation in FRIDA, together with the relatively small timestep of ½ of a year (Schoenberg et al., 2025a), causes the iterative solver to always converge after one iteration, if the hydrogen ion concentration of the previous timestep is used as the starting value. This simplifies the implementation of seawater carbonate chemistry in FRIDA compared to that in iLOSCAR.

The calculation of  $pCO_{2,ocean}$  is impacted by the temperature, salinity and alkalinity of the two surface ocean boxes. These six parameters are calculated as linear functions of the global mean surface temperature anomaly. This approach is based on

tests using emission-driven simulations of the MPI-ESM1.2-LR Earth system model (Mauritsen et al., 2019). The temperature of the high and low latitude surface ocean was thereby found to scale very well linearly with the global mean temperature anomaly; the scaling of the surface salinity and alkalinity is less linear, but still better than the often-used assumption of constant values (Figure S2). Temperatures drive changes in surface ocean salinity and alkalinity via melting of sea ice and glaciers in the high latitudes, which leads to the dilution of seawater. Hence, the temperature dependence is especially high in the cold surface box in FRIDA. In FRIDA-Clim default values at a global mean temperature anomaly of zero are taken from fits to MPI-ESM1.2-LR data, but a range of values are explored within the calibration for the temperature-dependent component (Table S1). In addition to the surface ocean properties that are relevant for the calculation of pCO<sub>2,ocean</sub>, a temperature dependence of the strength of the biological carbon pump in the cold surface ocean is also included, similarly based on MPI-ESM1.2-LR data and handled as the other dependencies described above.

# 505 2.6 Sea Level Rise




Sea-level rise (SLR) is included in the climate representation, as it is an additional metric of climate change that can lead to patterns of future climate impacts that are different from those of global mean temperature, especially under scenarios of stabilising or declining temperature levels, where the sea level would continue to rise. Global total mean SLR is modelled as the sum of five different contributions: thermal expansion of the ocean, the melting of mountain glaciers, the separate Greenland and Antarctic ice sheets, and changes in the storage of water on land. The full model setup is described in detail in Ramme et al., (2025; see appendix sections A1-A5 therein). Therefore, here only a short summary of the model components is given, focusing on the inputs from other parts of the model.

The thermal expansion of the ocean is modelled via a linear relationship to the change in heat content of the ocean (Marti et al., 2022). The ocean heat content change is calculated in the energy balance model of FRIDA-Clim and therefore directly available as input to this formulation.

The three SLR components that are linked to the melting of ice (mountain glaciers, and the Greenland and Antarctic ice sheets) are modelled as functions of the global mean temperature, which is provided by the energy balance model of FRIDA-Clim. The used relationships are taken from the literature and are discussed in more detail in Ramme et al., (2025).

The implementation of the final component of SLR - changes in land water storage - depends on the level of socioeconomic coupling. In a stand-alone version of the model, it can be modelled as a simple constant rate of SLR, as in other models (Nauels et al., 2017; Wong et al., 2017). In the Climate Module setup used in the FRIDA IAM, it is assumed that global population data is calculated internally, relating the annual SLR from land water storage changes linearly to the global population. In FRIDA-Clim, population input is therefore optional, with the simple constant rate used in its absence. Lastly, within the FRIDA IAM, agricultural and non-agricultural water withdrawal are modelled directly, and so is the installation





of hydropower plants. The water withdrawal component is translated into a total groundwater anomaly under the assumption that a fraction of water withdrawal is unsustainable and that this unsustainable fraction grows as the water demand increases. The total groundwater anomaly, which is assumed to be constantly refilling on a long timescale, is directly translated into a sea level anomaly. The hydropower component of land water storage changes is modelled as a linear function of the installed hydropower capacity (Schmitt and Rosa, 2024). The coefficient of this relationship is calibrated within the full model calibration of FRIDA (see Schoenberg et al., (2025) for details on the full model calibration).

# 2.7 Necessary Inputs to FRIDA-Clim and the FRIDAv2.1 Climate Module

The two models described here, while residing in a common overall framework, have different scopes and as such comprise a different subset of the climate system (Figure 1). The Climate Module in FRIDAv2.1 takes in just four exogenous timeseries, shown as grey hatched boxes in Figure 2: radiative forcings from solar output variations, volcanoes, and Montreal Gases, as well as the effect of Montreal Gases on stratospheric chemistry. All other anthropogenic climate drivers are simulated internally within the other FRIDA modules.

FRIDA-Clim, however, requires 18 further timeseries to be input. These are the seven anthropogenic climate drivers (black boxes in Figure 2) plus the 11 land use and crop inputs - nine land use transitions, plus crop production and forest cutting - described in Section 2.4. Additionally, population can be optionally input in order to model the land water storage component of SLR as discussed in Section 2.6.

# 545 3 Calibration and Initialisation

To ensure consistency with historical observations of key climate variables, FRIDA-Clim is calibrated against several observed datasets, largely based on the approach used to calibrate FaIR (Smith et al., 2024b). When utilised as the Climate Module within FRIDA, a two-step process is applied, wherein the Climate Module is first calibrated in a standalone manner - using a subset of the constraints used for FRIDA-Clim - and then further calibrated within the broader FRIDA IAM calibration process.

These procedures are visualised in Figure 4 and described here, with the FRIDA-Clim process detailed in Section 3.1, and the modifications for the Climate Module within FRIDAv2.1 in Section 3.2. The full set of parameters varied as part of the overall calibration is shown in Table S1, along with information on their representation in the calibration.

The emissions input to the historical ensembles in both model calibrations are ensured to be consistent with the calibration data for the full FRIDAv2.1 calibration over the overlapping timeperiod (1980-2022), but are necessarily extended back to 1750 to cover the full period. Only the total emissions of each species is input to the climate calibrations. The procedure for



the recent period is therefore set out in Table 1; extensions back in time followed the same sources where available, switching data sources when needed (e.g. PRIMAP-hist for extending CH<sub>4</sub> pre-1970) while ensuring harmonisation throughout the period. Land forcings - land use transitions, crop production, and forest cutting - are only applied in the FRIDA-Clim calibration, following historical Land Use Harmonization (LUH; Hurtt et al., 2019a, b, 2020) for transitions and cutting, and FAO crop data (FAO, 2023) from 1961, with the prior crop timeseries assumed to scale with population, itself taken from HYDE, (2025). See references in the Code Availability statement for full information on this procedure.

**First** Second FRIDA-Clim Constraint Constraint Air-sea **Priors GMST NPP** CO<sub>2</sub> Flux (30,000)**ECS TCR GMST Posteriors Passing** (100)(3,969)CO<sub>2</sub> Conc. **OHC** Aerosol ERF

Figure 4: Summary of the two-stage calibration process used within the modelling framework. Numbers indicate the ensemble sizes in the FRIDA-Clim calibration described in Section 3.1.

# 3.1 FRIDA-Clim calibration

The process for the calibration of FRIDA-Clim is adapted and extended from the FaIR calibration process (Smith et al., 2024b). As such, the model is calibrated in several steps, detailed below: a large prior ensemble is constructed and simulated; this is then filtered to remove members which do not approximately meet observations of some key variables; this filtered ensemble is then constrained to choose members which together roughly return the observed uncertainty distribution of broader climate variables and metrics. The Climate Module calibration (Section 3.2) was in practice performed first, with 10,000 spinup members and 100,000 priors, which was found to be far more than sufficient to generate a constrained ensemble; this fact, coupled with the greater model size of FRIDA-Clim, led to the use of a smaller ensemble in the FRIDA-Clim calibration.

# 3.1.1 Pre-industrial Spinup

There is uncertainty in many parameters within the ocean and land systems, which affects the equilibrium values of their carbon stocks (and consequently also ocean pH levels). Since the climate is considered to be in equilibrium in pre-industrial times, the land and ocean must be spun up to ensure an equilibrated climate to begin the historical period. A total of 31 parameters (nine in the ocean, 21 in the land, plus pre-industrial atmospheric CO<sub>2</sub> concentrations; see Table S1) are varied as part of this spinup procedure. This paper presents results and outputs from using 1,000 spinup members, but any number can be explored. The first calibration step is therefore to run the land-ocean model for 10,000 years, which was found to be sufficient to reach equilibrium in the carbon stocks, and correspondingly near-zero air-sea CO<sub>2</sub> flux. After this spinup, the 21 spinup stocks (15 on land plus the four ocean carbon boxes, and two ocean pH levels; see Sections 2.4 and 2.5) are output for each of the 1,000 members, and used as inputs to the historical calibration runs. Members of the spinup were only kept if their air-sea CO<sub>2</sub> flux magnitude at the end of the spinup period was less than 0.01 GtC/year - very small compared to a present-day flux of nearly 3 GtC/year and uncertainties in the global carbon budget (Friedlingstein et al., 2025).

#### 590 3.1.2 Prior Ensemble

With the equilibrium stocks generated for the 1,000 ocean parameter sets, a large historical (1750-2022) ensemble - here results are shown using 30,000 members – was performed, varying 72 parameters (Table S1). The accepted spinup parameter (and corresponding equilibrium stock) sets are repeated to generate the number of prior ensemble members required, as running one ocean spinup per prior member was considered prohibitively computationally expensive.








In the historical simulations, in addition to the spinup parameters, parameters were varied associated with the EBM, the forcing parameters (including the scaling factors (Section 2.2)), and parameters for the land and ocean relating to the sensitivity of changes to temperature and atmospheric CO<sub>2</sub> (which have no effect on the equilibrium stocks and were therefore not varied in the ocean spinup ensemble). The ocean parameters determining the surface salinity sensitivity to global temperatures in the warm and cold regions were set to co-vary with their alkalinity counterparts. See Table S1 for further details, and the Code Availability for the full information on the procedure.

#### 3.1.3 Posterior Constraints

The prior ensemble is then constrained in a similar way to the two-step process used for FaIR (Smith et al., 2024b); see Figure 4. In the first step, rough consistency with key variables over the whole historical period is enforced, while in the second, a joint constraint is applied to ensure the uncertainty distributions across a range of climate variables match the best-estimate present-day observations as closely as possible.

The first constraint excludes all runs which have a mean root-mean squared error (RMSE) over the period 1850-2022 greater than 0.16K (the approximate uncertainty in present-day observations) for temperature when compared to observations, as applied in FaIR's calibration. Additionally, the ocean component allows for a constraint on observations of the air-sea CO<sub>2</sub> flux, with runs excluded if they have an RMSE over 1960-2022 which is greater than 20% of the mean observed value over this period (observations from the global carbon project (Friedlingstein et al., 2025), equating to a value of 0.39 GtC/year). Finally, the total NPP is filtered to exclude members with a year 2000 value more than 10GtC away from the observed result of 59.22 GtC in Haberl et al., (2007). These three constraints together reduce the 30,000 member prior ensemble to 3,969 (Figure S3).

In the second constraint step, the distributions of key climate quantities were constrained to closely match observational and IPCC AR6-assessed ranges. This was applied simultaneously to total aerosol forcing (averaged 2005-2014), Equilibrium Climate Sensitivity (ECS), Transient Climate Response (TCR), Ocean Heat Content (OHC; 1971-2020 change), CO<sub>2</sub> concentration (2022), and GMST anomaly (2003-2022 relative to 1850-1900). The aerosol forcing, ECS, TCR, and OHC constraints are taken from AR6 (Forster et al., 2021), with the CO<sub>2</sub> concentration and GMST constraints updated to the recent Indicators of Global Climate Change (IGCC) data which updates IPCC Working Group 1 assessments (Smith et al., 2024a). A sample of 100 parameter sets was taken from this sample to generate the posterior ensemble; see Figure S4 for the prior, constrained, and observational distributions of these quantities.






These 100 parameter sets in FRIDA-Clim, comprising 72 parameters and 21 resultant stocks, sampling uncertainty across the land, ocean, and atmosphere, can be used to simulate the climate and carbon cycle response to various future scenarios (Section 4).

#### 3.2 FRIDAy2.1 Climate Module calibration

The Climate Module of FRIDAv2.1 is calibrated separately from - and prior to - the calibration of the full IAM, for multiple reasons. Firstly, the strong interdependence between climate parameters (see Smith et al., (2024b)) necessitates the creation of climate parameter sets, which is not possible under the full model calibration in which all parameters are varied separately (Schoenberg et al., 2025a). Secondly, the IAM is initialised in 1980, but components of the climate system feature uncertainty throughout the historical period, necessitating the need to calibrate over the whole period from the pre-industrial to the present day and provide initial condition sets from which to run FRIDA from 1980. Thirdly, the variation of the equilibrium pre-industrial ocean carbon stocks necessitates a separate climate-only spinup.

The procedure for calibrating the Climate Module of FRIDAv2.1 is conceptually similar to that of FRIDA-Clim, with some key differences. The main difference is the lack of a representation of land in the Climate Module, resulting in fewer parameters being varied and necessitating that the historical priors be forced with observed land variables.

Due to the reduced scope of the Climate Module, a bespoke ocean spinup is ran (10,000 here by default), in which only 10 parameters are varied (nine in the ocean, plus pre-industrial atmospheric CO<sub>2</sub> concentrations; see Table S1), affecting the six ocean stocks (four carbon, two pH). The same air-sea CO<sub>2</sub> flux filtering as in FRIDA-Clim is applied, and the resultant ensemble is again repeated in order to run the large prior ensemble, with 100,000 members by default.

The prior ensemble is driven by the same anthropogenic emissions, but the land component of the carbon cycle is instead driven using an exogenous timeseries for the terrestrial carbon balance (see Section 2.4) from Friedlingstein et al., (2025). A total of 44 parameters are varied within this ensemble (Table S1).

The prior ensemble is constrained identically to that of FRIDA-Clim (Figure 4), with the exception of the NPP constraint which cannot be used for the Climate Module since NPP is not simulated here. The resultant posterior sample members (100 by default) are then taken to input as distinct climate parameter sets to FRIDAv2.1, allowing for the systematic exploration of the climate uncertainty across key variables in the FRIDA IAM. In all, 13 parameter set-dependent stocks in year 1980 are additionally provided to the full FRIDAv2.1, to complete the calibration and initialisation process; these are the ocean stocks (two pHs and four carbon stocks), the three energy balance model temperatures, the initial changes in ocean heat and atmospheric CH<sub>4</sub> and CO<sub>2</sub> (the latter two affected by temperature via the lifetime feedback and hence calibration-dependent),



and finally the surface temperature offset from 1850-1900 relative to 1750. This latter stock varies due to parameter-dependent variations in the temperature response in the early period, rendering a varying non-zero 1850-1900 temperature relative to the (initialised at zero) 1750 level, and is required in order to express the surface temperature in a policy-relevant way.

Following the Climate Module calibration, the full model calibration (Schoenberg et al., 2025a) ensures FRIDA is consistent with observational data across the model structure along the FRIDA calibration timeperiod 1980-2023 - i.e. from model initialisation to the present day. Several FRIDA calibration timeseries reside in the climate module; these are listed for completeness in Table S3, along with their data source information. The total emissions timeseries are ensured to be identical to those used in the climate module calibration, ensuring consistency across the different calibration procedures.

# 4. Experiments using FRIDA-Clim

Using the calibrated parameter sets, the standalone FRIDA-Clim simple climate model and the coupled FRIDAv2.1 IAM can be used to explore experiments of interest. As a coupled IAM, FRIDAv2.1 is suitable for the exploration of future scenarios, the process of which is described in Schoenberg et al., (2025) and will be explored further elsewhere. As a simple climate model, FRIDA-Clim can be used to study both future scenarios and idealised experiments, as briefly explored here using the 100-member calibrated ensemble.

# 4.1 Idealised CO<sub>2</sub> Experiments

Four idealised CO<sub>2</sub> emissions-based experiments have been suggested for use in model intercomparison for both ESMs and SCMs, termed flat10MIP (Sanderson et al., 2023, 2025). The response of FRIDA-Clim to these experiments is shown in Figure 5. Each experiment features constant 10GtC/yr CO<sub>2</sub> emissions (similar to present-day levels) for 100 years, with several experiments then branching off (panel a). The input of carbon to the atmosphere leads to substantial uptake in both the ocean and land, with the latter sink weakening over time (e). Warming in year 100, after 1000GtC cumulative emissions, is termed the Transient Climate Response to Emissions (TCRE), and is around 1.5°C in FRIDA-Clim (panel i).



Figure 5: Response of FRIDA-Clim to idealised emissions scenarios. Shown are the emissions trajectories (a), GMST response as a function of time (b) and cumulative emissions (c), and carbon sink responses for the distribution of flat10-cdr as a function of cumulative emissions (d) and median timeseries for all scenarios (e-h). Also shown are distributions of key metrics: T100yr, the warming in years after 1000GtC emissions in flat10 (i); the temperature change after 50, 100, 300 years of ceasing emissions, termed the zero emissions commitment (ZEC; j); the difference between the year of peak warming and the year of net zero emissions in flat10-cdr (t-PW; k); and the warming in years 150/200/310 of flat10-cdr minus that in 125/100/0 in flat10, representing variations from a linear response (termed TNZ, T0, T1000; l). All metrics are calculated using 20-year means centred on the years noted.

The next idealised experiment is designed to study the Zero Emissions Commitment (ZEC), defined as the additional temperature change upon cessation of emissions, denoted ZECx when calculated x years after emissions stop. In the flat10-




695 zec experiment, emissions are instantly set to zero in year 100, with FRIDA-Clim demonstrating progressively more negative median values (i.e. global cooling) of ZEC50, ZEC100, and ZEC300 (j). This is ultimately driven by the continued flow of carbon from the atmosphere into the land and ocean (f).

The flat10-cdr experiment features a linear drop in emissions from year 100 to -10GtC/yr after a further 100 years, continuing until cumulative emissions reach zero again after 300 years; this allows for exploration of the reversibility of the climate system. In this, FRIDA-Clim shows near overall reversibility in the land sink in this scenario, with the ocean retaining a substantial amount of carbon (d), and atmospheric concentrations consequently lower than in year zero upon emissions removal. This also manifests in peak warming occurring before cumulative emissions hit zero (t-PW; k).

Finally, the emissions trajectory in flat10-nz follows flat10-cdr but settles at zero emissions rather than going negative, to study the effect of net zero emissions reached via a linear emissions reduction. The carbon stocks ultimately stabilise similarly to flat10-zec (h), with GMST declining but at a higher level, due to the greater cumulative emissions.

Global temperature reversibility is further tested by the calculation of the response at the same level of cumulative emissions rach net zero in flat10-cdr at 1250GtC cumulative emissions minus the response in esm-flat10 under the same total; TR1000, which subtracts the temperature in flat10-cdr in year 200, upon reaching 1000GtC cumulative emissions via emissions removals, from that in flat10 at the same cumulative emissions; and TR0, the temperature in flat10-cdr after full emissions removal. All three metrics are negative in the median in FRIDA-Clim (panel 1), with more negative values under emissions removals.

FRIDA-Clim thus simulates approximate linearity in the land carbon response under idealised removal, with hysteresis in the ocean sink and consequent over-compensation in the atmospheric response. Values of temperature reversibility metrics (ZECx, TNZ, TRx, t-PW) are therefore negative in the median. Generally, these results are consistent with the findings across flat10MIP (Sanderson et al., 2025), which found similar carbon budget responses. The ZEC and TR metrics are more negative than most flat10MIP models, with a response seemingly close to that in the CICERO SCM; t-PW in FRIDA-Clim is more negative.

# **4.2 Future Emissions Scenarios**

Figure 6 shows the historical and future trajectories of surface temperature (top) and carbon cycle components (bottom) under four shared socioeconomic pathway (SSP) scenarios: ssp119 (strong mitigation), ssp245 (medium emissions), ssp534-over (strong overshoot) and ssp585 (very high emissions) (O'Neill et al., 2016; Tebaldi et al., 2021). Results are shown for



both FRIDA-Clim and the FaIR SCM; to ensure consistency between the comparisons, FRIDA-Clim was calibrated here to use RCMIP historical emissions as applied in the FaIR data used. The external time series of greenhouse gas and aerosol emissions are extended beyond 2100 following Meinshausen et al., (2020). The SSP scenarios branch off the historical run in 2015.

Figure 6: Response of GMST (top) and carbon cycle (bottom) in four scenarios in FRIDA-Clim. GMST values are also shown for an 841-member FaIR ensemble calibrated using the same emissions. The carbon cycle response displays median fluxes into (positive) or out of (negative) the land, ocean, and atmosphere. Median values and 16-84 and 5-95 percentile ranges are shown for GMST. Observational GMST data are shown in black (Smith et al., 2024a). The individual time series of the atmospheric CO<sub>2</sub> concentration, as well as ocean and land carbon uptake, including their percentile ranges, are shown in appendix Figure S5.





Due to the model calibration, the observed temperature increase in the historical period is matched well by FRIDA-Clim, with a small spread between the ensemble members. The following temperature evolution is very different between the scenarios, consistent with the prescribed emissions pathways. The long-term temperatures relative to pre-industrial times stabilise under each scenario at very different levels - from under 1°C in ssp119 to around 6°C in the median for ssp585. The scenarios with mitigated warming (ssp119 and ssp534-over) are the first to reach a stable climate around the second half of the 22nd century, although both show a minor long term cooling trend after that. The ssp585 scenario stabilises around 2250, while ssp245 is relatively stable already in 2100, but then exhibits a long term cooling trend that is stronger than in the mitigation scenarios.

FRIDA-Clim projects a sharper initial temperature increase than FaIR in the 21st century under high emissions, with subsequent faster turnaround and decline of temperatures after peak warming is reached. Both of these model differences between FRIDA-Clim and FaIR can be seen in the evolution of the atmospheric CO<sub>2</sub> concentration (see Figure S5), indicating that the new process-based carbon cycle in FRIDA-Clim is more responsive to CO<sub>2</sub> emissions than the impulse-response model in FaIR.

Furthermore, FRIDA-Clim projects long-term temperature anomalies that are lower than those from FaIR by roughly 0.5K or more in all scenarios. These long-term differences in the temperature anomaly between FRIDA-Clim and FaIR are qualitatively consistent with the atmospheric CO<sub>2</sub> response (Figure S5), though in ssp585 the discrepancy in temperature appears larger, suggesting slightly different climate sensitivities or temperature feedbacks to other species in the calibrated ensembles.

The new carbon cycle in FRIDA-Clim allows for a more detailed look into the sequestration of carbon into the land and ocean reservoirs (Figure 6). The land becomes a net sink of carbon only in the late 20th century, because emissions from the food and land use sector outweigh the natural land sink before that. The uptake of carbon by the land then quickly increases and overtakes that of the ocean. The further evolution of the carbon cycle is strongly scenario-dependent. When emissions increase strongly, the share of CO<sub>2</sub> that remains in the atmosphere increases. The scenarios with continuous warming (ssp245, ssp585) show that the land sink weakens more sharply than the ocean sink under falling emissions after 2100, and diminishes around 2200, while the ocean keeps taking up carbon until the end of the simulation period at a decreasing rate.

In the strong mitigation scenarios (ssp119, ssp534-over), the ocean turns to a sink shortly after the CO<sub>2</sub> emissions become negative, while the land follows a few decades later. In these scenarios, the ocean and the land reach an equilibrium, and CO<sub>2</sub> fluxes approach zero around 2200, almost immediately after the negative emissions cease. In the early phase of negative emissions, the ocean and land are still carbon sinks, so that the atmosphere loses more CO<sub>2</sub> annually than the component from negative emissions.

#### 5. Discussion






The radiative forcing components simulated within the FRIDA framework represent key contributors of the total, including key GHGs such as CO<sub>2</sub>, CH<sub>4</sub>, and N<sub>2</sub>O, as well as other important forcers such as aerosols, plus chemical effects on ozone and stratospheric water vapour. Twelve distinct radiative forcings are included, nine simulated interactively and three imposed exogenously - including two (volcanic and solar) natural forcings. Together, these comprise a broad set of climate forcers, but still a simplification compared to the set present in the FaIR model, on which many of these forcings are based. This simplification was necessary within the Climate Module to allow for the drivers of these forcers to be simulated interactively at the process-level within FRIDAv2.1, allowing for the incorporation of the feedback loop between anthropogenic emissions, the climate response, and subsequent impacts on the sources of emissions. FRIDA-Clim, as a standalone model, could incorporate additional species, but its current form lends itself to serving as a plug-in climate module for other modelling frameworks. Despite the reduced number of forcers compared to FaIR, FRIDA-Clim reproduces observed climate variables well throughout the calibration process.

Only one aerosol species -  $SO_2$  - is simulated. This was deemed an appropriate simplification, as other aerosol species are typically co-emitted with  $SO_2$ , or at least follow similar global trajectories, and also because the aerosol forcing response is calibrated to the present-day estimate. Additional aerosol species could be incorporated as part of future model development, especially if a focus on sources of carbonaceous aerosol such as fires is undertaken.

Even when coupled within FRIDAv2.1, three radiative forcings are externally imposed on the Climate Module, with no feedback possible within the system. Two of these are natural forcings - volcanic and solar - for which no feedback would need to be simulated. However, the effects of Montreal Protocol-controlled gases - via their direct forcing and their effect on ozone - are assumed to steadily decline over time, in keeping with international agreements regulating their decreased use. However, there is scenario dependence in some of these species within the SSPs, reflecting some uncertainty in this, the dynamics of which could plausibly be modelled within FRIDA, and will be considered for future development.

Both FRIDA-Clim and the Climate Module are calibrated to ensure consistency with historical temperatures and air-sea CO<sub>2</sub> fluxes, as well as present-day best-estimates of aerosol forcings, temperature, CO<sub>2</sub> concentration, and OHC change, and the intrinsic ECS and TCR climate properties. In addition, FRIDA-Clim is constrained on observed NPP data, and checked for agreement with observations of the terrestrial carbon balance and land carbon sink. Because of the lightweight nature of this calibration, these inputs can be easily updated and the model recalibrated as new data becomes available, as can the emissions timeseries used as inputs to the calibration. The default sizes of the spinup, prior, and posterior ensembles can also be modified as desired.





While the Climate Module is calibrated to reproduce present-day climate variables when forced with historical emissions, the full FRIDA IAM is initialised in 1980, after which it is free-running. Deviations from the best-estimate historical emissions timeseries are therefore inevitable, especially within the context of the large ensemble uncertainty exploration which FRIDA is designed for (Schoenberg et al., 2025a). The calibration of the full model ensures it closely reproduces these best-estimate emissions, but a variation in the present-day distribution of climate properties from the calibration constraints will still be present.

The EBM parameters in the prior ensemble are calibrated on ESM data using abrupt-4xCO<sub>2</sub> experiments, and as such capture the climate feedbacks simulated within these complex climate models. However, the model is used to simulate temperatures above suggested thresholds for activating "tipping points" within the Earth system (McKay et al., 2022), which were not incorporated in FRIDAv2.1 due to the substantial uncertainties present, as well as the regional dependence of their expected impacts. Future development and use of the FRIDA IAM will explore the incorporation of these effects in an idealised manner.

Some other areas remain for future development, in addition to those noted above. The stratospheric H<sub>2</sub>O forcing can be driven by the chemically-relevant concentration of CH<sub>4</sub> rather than its forcing; the small degraded land area can be incorporated into the albedo estimate; and the prior ocean parameters could be further constrained based on observations or multi-model data.

#### 6. Conclusions

FRIDA-Clim has been developed and presented here as a standalone simple climate model oriented in the system dynamics framing. The radiative forcing and energy balance response in FRIDA-Clim is based on the widely-used FaIR simple climate model (Leach et al., 2021). In addition, a process-based carbon cycle model was implemented that allows the tracking of the sequestration of carbon into the land and ocean reservoirs. Apart from its use as a standalone simple climate model, FRIDA-Clim can also be integrated into IAMs; its integration as the Climate Module of the FRIDAv2.1 IAM has been documented here.

As a standalone climate model, FRIDA-Clim takes anthropogenic drivers - across emissions and land use - along with natural forcings and computes their effect on the climate system, via their influence on radiative forcing and CO<sub>2</sub> concentrations. When integrated as the Climate Module within the FRIDAv2.1 IAM, it takes almost all these climate-relevant outputs from the other modules. Extensive connections are made to other modules within FRIDAv2.1, with emissions and other climate drivers originating from all but one other module, and detailed interactions within the land use

https://doi.org/10.5194/egusphere-2025-4766 Preprint. Discussion started: 4 November 2025

© Author(s) 2025. CC BY 4.0 License.

845

850

855

EGUsphere Preprint repository

domain. Both versions are fully calibrated to reproduce observations of key climate variables, with a systematic exploration

of uncertainty in parameters relating to the radiative response, energy balance model, and the ocean and land carbon cycles.

FRIDA-Clim produces temperature projections in the SSP context that are similar to those of FaIR, although some

qualitative differences exist. Similarly, FRIDA-Clim's behaviour is comparable to that of more complex Earth system

models and other simple climate models in the setting of the idealised flat10mip CO2 emissions experiments. The inclusion

of a process-based carbon cycle in FRIDA-Clim is a useful feature in light of a general drive towards more process-based

and emission-driven models (Sanderson et al., 2023).

The incorporation of this simple climate model systematically, and in an integrated manner, within the broader FRIDAv2.1

framework allows for the exploration of coupled dynamics and uncertainties across the human-Earth system, across a range

of future scenarios. The provision of the new, lightweight simple climate model FRIDA-Clim v1.0.0, with an integrated

carbon cycle and reflecting key uncertainties, extends the set of such models and as such aids in the exploration of structural

uncertainty within multi-model frameworks such as RCMIP (Nicholls et al., 2021, 2020).

**Code Availability** 

The FRIDA-Clim calibrations and SSP and flat10 setups, including code for all processing and data figures in this paper, are

at https://zenodo.org/records/17207036 (Wells & Ramme, 2025); data for the RCMIP calibration are used for Figures 5 and

5 (see the readme file and scripts). The calibration procedure for the Climate Module is a

https://zenodo.org/records/17207019 (Wells et al., 2025). Version 2.1 of the full FRIDA model can be found at

https://zenodo.org/records/15310860 (Schoenberg et al., 2025b), with information on the process for the creation of the

emissions, forcings, concentrations, and temperature for use in both calibrations - of FRIDA-Clim v1.0.0, the Climate

Module, and then of FRIDAv2.1 – contained within.

**Author Contribution** 

Conceptualisation: all authors. Funding acquisition: WAS, CS, CL, CM. Investigation: CS, CW, LR, JB, AM, WAS.

Methodology: CS, CW, LR, JB, AM, WAS. Writing - original draft: CW, LR. Writing and editing: all authors.

**Competing Interests** 

The authors declare that they have no conflict of interest.

33

# 865 Acknowledgements

This research was supported by the Horizon Europe research and innovation programs under grant agreement no. 101081661 (WorldTrans). This work used resources of the Deutsches Klimarechenzentrum (DKRZ) granted by its Scientific Steering Committee (WLA) under project ID 33.

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
