# Peer review of "FRIDA-Clim v1.0.0: a Simple Climate Model with Process-Based Carbon Cycle used in the FRIDAv2.1 IAM"

_EGUsphere, 2025_

## Author Comment (AC1)

Many thanks to the reviewers for this useful feedback. We have incorporated these into the paper. Specifically, we have clarified some key arguments and motivations, adjusted the model description figures, and have carried out further analysis to justify the choice of regression-based forcers and explore the differences between FRIDA-Clim and FaIR's temperature responses to the SSPs. We have updated the Zenodo accordingly, and also therefore iterated the model version to v1.0.1. Detailed responses are given below in blue, with quoted changes in italics; line numbers refer to those in the tracked changes version.

**Reviewer 1**

**1. General comments**

The authors present relevant extensions to the FRIDA framework. The climate impact of Green House Gases such as CO2, CH4, N2O, aerosols, stratospheric water vapor can be simulated, and together with the FRIDA IAM, the dynamics of the coupled human-Earth system can be modeled. Idealized CO2 emission experiments are performed and detailed calibration routines and explained in this work. The paper is very comprehensive and detailed which probably is due to the complexity of the involved models and quantity of implemented functionalities.

Thank you very much for your thorough and thoughtful review and overall positive comments.

The authors try to describe the uniqueness of FRIDA-Clim, however, these arguments should be further elaborated: In which key aspects differs FRIDA-Clim from other/previous models? What are the unique selling points? Furthermore, the philosophy of the model should be better motivated: Why do the authors choose a "minimal required climate" approach?

The "minimal required climate" approach is a general philosophy that extends beyond FRIDA as a use case for coupling climate modules to integrated assessment models. For example, the IPCC Climate Assessment package (including FaIR as one of its three climate models) incorporates around 50 emissions species, many of which are minor fluorinated greenhouse gases that are not individually modelled by IAMs as they do not have the requisite sectoral detail. Therefore, we a looking to strike a balance between a detailed enough coverage of all climatically important emissions species with the sectors and emissions species that are modelled in IAMs, which may have different sectoral breakdowns than FRIDA. We have given some more detail on this below including paper edits.

The authors try to explain the interplay and interfaces of the different modules. A major issue is that the used terminology is not always well defined nor used within the text. Important key words such as external, internal, endogenous, coupled, uncoupled, integrated should be well defined and not mixed within the text. This would better describe the boundaries and interfaces of the different modules and the reader would better understand which component is meant.

Thank you for this observation. In the revision, we hope that we have better defined some of these terms better.

**2. Specific comments**

**Title**

- Use of abbreviation "IAM" in the title not recommended, better write the full wording "Integrated Assessment Model".

We agree with this and have made this change.

- In the beginning, it is not clear to the reader, that these are two software modules: FRIDA-Clim and FRIDA IAM. This is later explained in the abstract, but the title should be more self-explanatory, otherwise it is confusing to the reader having two version numbers.

- Maybe a different terminology would help, such as FRIDA framework or FRIDA "core" for the overall framework, and FRIDA-Clim as extension/module to that framework?

We have grappled with how to frame this issue within the project, and we feel that using the "IAM" term to refer to FRIDAv2.1 best describes its orientation in the literature. Similarly, FRIDA-Clim is slightly different to the Climate Module in FRIDAv2.1 as documented here (though they are being brought into closer alignment under model development), so it would not be possible to refer to it as an extension here. Instead to address this issue we have moved the term "Integrated Assessment Model" before FRIDAv2.1, to emphasise the nature of this model and therefore its distinction from the Simple Climate Model.

**Abstract**

- What is the other part of the *two-way feedback*?

This refers to the feedbacks between the two components of the human-Earth system modelled by FRIDA; one part is the effect of humans on the climate - as modelled by FRIDA-Clim - and the other is the effect of the climate on humans, i.e. climate impacts. We have clarified this sentence to note this now (L23-4): *"Connecting anthropogenic emissions to the resultant climate response is one part of this the two-way feedback within this system, with the resultant climate impacts the other."*

- Add definition of *coupled* and *uncoupled* . Otherwise, it is unclear to the reader which software module couples to what other module.

We have modified the sentence on L30-2 to clarify this point: *"In both uses, i.e. with the climate response interactively connected to the upstream human drivers and downstream climate impacts within the FRIDA IAM (coupled) and when ran separately as FRIDA-Clim driven by exogenous forcings (uncoupled), its climate drivers are simplified as compared to FaIR"*

**Introduction**

- Both FRIDA-Clim and the integrated Climate Module are documented here

Yes; we have now clarified the abstract to better set this out (L25-6): *"This paper documents both the Climate Module within FRIDAv2.1, and the modified version separately simulated as a standalone simple climate model termed FRIDA-Clim version 1.0.1."*. This better sets the reader up for the confirmation of this point on L82-3 (*"Both FRIDA-Clim and the coupled Climate Module are documented here."*).

- What it meant by the *integrated* Climate Module? Is it related to *coupled* mode as stated in the abstract?

Yes, this refers to the coupled setup; we have replaced "integrated" with "coupled" here to aid consistency and clarity.

**Model description**

- In the text, the terms *external*, *internal*, *endogenous* should be better explained. How do these modes relate to coupled/uncoupled/integrated ?

We regret the lack of clarity on this point in the original version. We have added a sentence on L86-8 in the introduction to explicitly set out which terms refer to which model (the Climate Module or FRIDA-Clim):

*"Since FRIDA-Clim is run uncoupled from the rest of the IAM, it must be driven with external, exogenous drivers (e.g emissions and land use changes), whereas the Climate Module integrated in the FRIDA IAM receives drivers modelled interactively within the IAM, as internal endogenous climate forcings."*

We feel that this now sets up the reader to understand the uses of "external", "endogenous" etc when these are used in the model description.

- Philosophy should be better motivated

We have added to the paragraph on L138-9 now to more clearly motivate this; it now reads:

*The philosophy of FRIDA-Clim, in keeping with that of FRIDAv2.1 as a whole, is that only the minimum level of detail in the climate system required in order to adequately reproduce historical and future expected global climate dynamics should be represented. This approach stems from FRIDAv2.1's setting within the System Dynamics approach (Schoenberg et al., 2025a), in which only the key components of a given system are simulated in order to better focus on the feedbacks between these components. As well as aiding legibility, this ensures a quick model runtime, allowing for the role of parameter uncertainty to be explored in depth.*

- Why should climate functionality minimized? e.g. Model performance reasons?

This is a good point, as we should have detailed it further in the manuscript. Model runtime is a key reason here, in addition to model legibility. We have now added these sentences on L140-3 to detail this: *"This approach stems from FRIDAv2.1's setting within the System Dynamics approach (Schoenberg et al., 2025a), in which only the key components of a given system are simulated in order to better focus on the feedbacks between these components. As well as aiding legibility, this ensures a quick model runtime, allowing for the role of parameter uncertainty to be explored in depth."*

- Table 1 give a very good overview of anthropogenic climate drivers

Thank you for your positive comment.

- Section 2.2 Effective Radiative Forcing: Motivate why ERF concept is used instead of RF concept.

We use ERF to account for the fast response to instantaneous forcings; we have now added this sentence on L210-2 to motivate this approach: *"Forcings are calculated as ERFs in order to incorporate the fast feedbacks that differentiate ERF from the instantaneous forcing (Forster et al., 2016), as is standard in simple climate models (e.g. Nicholls et al., 2021)."*

- Section 2.2.4 Aerosols: FRIDA or FRIDA IAM?

We note at the start of Section 2.2 that these ERF calculations are the same for both versions: *"The simulation of Effective Radiative Forcings (ERFs) is performed identically in both FRIDA-Clim and the FRIDAv2.1 Climate Module"* - we feel this is a natural place for readers to refer back to for this information from further subsections, for clarity.

- Section 2.2.5 Ozone: Is FRIDA and FRIDA-Clim focusing on effects within troposphere/stratosphere or both? Which of the anthropocentric emissions have the largest contribution to ozone increase or decrease?

Following the recommendation of the IPCC Sixth Assessment Report Working Group 1 Chapter 7, we do not separate the forcing into these components; the total is simulated. This also follows the treatment in FaIR. We have added this sentence to clarify this on L256-7*: "Tropospheric and stratospheric ozone forcings are not disaggregated here, with the calibration trained on the combined effect."*

In a full climate model, the largest contribution to ozone forcing would be from NOx emissions (Smith et al., IPCC AR6 WG1 Chapter 7 SM, table 7.SM.3). Since we do not model NOx emissions exogenously and they are based on N2O emissions (for the non-AFOLU component) and SO2 emissions (for the AFOLU component), and CO and NVMOC both scale with methane emissions in our regression-based relationship as well as methane itself being a major factor in ozone formation, methane emissions has the largest contribution to ozone forcing in FRIDA-Clim.

- Section 2.2.9 No reference given for stated approach of linear relation between CH4 and stratospheric water forcing

Thank you for highlighting this oversight. We have added to this section to note the origin of this as from the FaIR model (L312-3): *", as simulated in the FaIR model (Leach et al., 2021)."*

- Section 2.2.11 Volcanic: Which are the concrete emissions linked to volcanic activity driving the resulting forcing?

Volcanic forcing is driven primarily by sulfur aerosols - we have added to the first sentence of this subsection (L321-2) to briefly note this: *"The first exogenous natural forcing represented in FRIDA is the negative forcing due to volcanic eruptions, primarily from sulfur emissions."*

- Section 2.4: between FRIDA-Clim and the Climate Module

   - Are these different modules? To my understanding, FRIDA-Clim is the climate module. Do you mean the "core" climate module of FRIDA IAM?

Apologies for the lack of clarity on this point here. FRIDA-Clim is the standalone simple climate model, while Climate Module refers to the module representing the climate in the FRIDA IAM. We hope the clarifying additions to the introduction aid in this, and we have further clarified this here in this sentence (L358-9): *"The land carbon cycle represents the domain with the largest*

*differences in approaches between the standalone FRIDA-Clim SCM and the Climate Module coupled within the FRIDA IAM,".*

- Section 2.4.1 Units GtC Mha-1 yr-1

  - Hectar is a non-SI unit. Unless this a very common unit in this research field or context, I would suggest to convert into SI-units, e.g. km$^2$ or m$^2$

We use data from FAO to inform and calibrate this, which uses Mha units (https://www.fao.org/faostat/);

- Section 2.4.2 Soil Carbon, equation (9) mix of units °C and K in same equation should be avoided!

We have adjusted this to be °C only.

**Calibration and Initialization**

Section 3.1 FRIDA-Clim calibration

- Missing motivation for relevance of ensemble sizes, members. see also given numbers in Figure 4. What does it mean to have an ensemble size of 30,000? Is this a large or small number? How does this number compare to?

We agree this needs explaining within the manuscript. This number of priors was found to give the desired number of posterior members (100), which we have now clarified (L630-2): *"This number was found to be sufficient to result in over 100 members after constraining, which was deemed a suitable balance between computational runtime and uncertainty exploration."*

**Experiments using FRIDA-Clim**

- Section 4.1 Idealized CO2 Experiments

- Figure 5. panels i,j,k,l missing description/units of y-axis

We have now added the units (probability density) to these subplots.

- Better explain and motivate used metrics ZEC, TNZ, T0 etc. Consider putting definitions of and motivations for used metrics into a table for a better overview.

We have added a new paragraph to better motivate and define these metrics, which now reads (L716) *"These are designed to study the behaviour of the climate system, including the carbon cycle, under reversibility of CO2 emissions, in order to inform the understanding of more realistic emissions scenarios. Several important metrics can be calculated from these experiments (Sanderson et al., 2025):..."* and then defines the metrics before detailing the scenarios. The TNZ, TR0, TR1000 metrics require the scenarios to be defined first, so are explained fully in the Figure 5 caption still. The figure 5 caption has been shortened to remove duplicate explanation, and the subsequent paragraphs also clarified.

- Last paragraph concludes: *FRIDA-Clim thus simulates approximate linearity in the land carbon response under idealised removal, with hysteresis in the ocean sink and consequent

over-compensation in the atmospheric response.... Generally, these results are consistent with the findings across flat10MIP (Sanderson et al., 2025)*

   - Explain better in the individual graphs, how authors came to these conclusions.

We have added to this paragraph (L769) to explain this and refer to the panels in Figure 5 which ground these conclusions, and also to the figures in the flat10-mip Sanderson et al 2025 paper which present these metrics across other models. The paragraph now reads:

*FRIDA-Clim thus simulates approximate linearity in the land carbon response under idealised removal, as cumulative land carbon uptake approaches zero as cumulative emissions reach zero. Conversely, hysteresis occurs in the ocean sink (positive uptake as cumulative emissions reach zero) with consequent over-compensation in the atmospheric response (Figure 5d). Values of temperature reversibility metrics (ZECx, TNZ, TRx, t-PW) are therefore negative in the median (Figure 5j-l). Generally, these results are consistent with the findings across flat10MIP (Sanderson et al., 2025), which found similar carbon budget responses (their Figures 5, 13).*

3. Technical corrections

- Consider introducing abbreviations of other models as well: FaIR, DICE, LPJmL, LOSCAR, iLOSCAR, CICERO, ...

We have introduced these acronyms now where each model is first mentioned in the main text, with the exception of CICERO which we only refer to briefly once in reference to Sanderson et al 2025, which motivated readers can refer to for further information.

- Introduce abbreviation AFOLU

Done

- Section 2.2.7 Introduce abbreviation GHG

Done

- Section 2.3: Introduce abbreviation GMST

Done

- Figure 1 gives a very good overview of the dependencies of the different software modules. However, the chosen colors of the lines are not ideal for people with red-green deficit. Consider use of different line styles, e.g. continuous and dashed.  Abbreviation *EBM* in the figure caption is not introduced.

We have used different linestyles here to mitigate any issues. We have fixed the abbreviation issue.

- Figure 2: revise colors of lines, especially orange/red-green lines

We checked this figure using an online simulator and found it was understandable to each type of colour blindness (https://www.color-blindness.com/coblis-color-blindness-simulator/). Unlike Figure 1, the line colours in Figure 2 don't affect the figure's understanding - the figure is legible in greyscale too. The boxes already use different hatching styles to enable this. Also, the current module colourscheme has been defined project-wise, and is used in the published

overview paper (https://gmd.copernicus.org/articles/18/8047/2025/). For these reasons, we would prefer to keep the figure colours as they currently are.

- Figure 3: revise orange/red - green colors. Colors might be hard to distinguish for some people. The arrows under the "fast soil" and "slow soil" boxes are confusing. Is there a way to summarize/aggregate/abstract the exchanges between the different components for a clearer visual representation?

We have adjusted the colour of the soil respiration lines, and again have checked for reasonable interpretability for colour blindness. The current colours mostly reflect common understandings of the colour of the process involved (eg blue for ocean, green for forest, brown for soils) and we feel this is a reasonable overall approach. We agree the effect of land transitions on soil carbon is complicated by the number of processes, but were unable to find a better approach to show this. We instead now refer to the relevant section in the caption for the detail.

- What is meant by (t-PW; k) Is this a reference to one of the figure panels? Then, it is better to explicitly write "see Fig. 5, panel (k) ...", same comment applies to similar used abbreviations in this section.

We agree this was unclear before; this specific instance has been altered by the changes made above.

**Reviewer 2**

This manuscript introduces FRIDA-Clim v1.0.0, a Simple Climate Model (SCM) designed for use both as a standalone tool and as the integrated Climate Module within the FRIDAv2.1 Integrated Assessment Model (IAM). The model differentiates itself from standard impulse response SCMs (such as FaIR) by incorporating a process-based carbon cycle for the ocean and land. The authors detail the model structure, the coupling between human and natural systems, and a rigorous two-stage calibration process involving a 30,000-member prior ensemble constrained against historical observations. The model is evaluated against idealized $CO_2$ experiments (flat10MIP) and standard SSP scenarios, demonstrating general consistency with observational records but notable divergence from the FaIR model in high-emission future scenarios.

I suggest the following issues be addressed by authors before publication.

1- The model simulates emissions of NOx, VOCs, CO, and Black Carbon via linear regression against other drivers (such as N2O and SO2). While historically valid, these correlations are structural weaknesses in an IAM context, as future policies may decouple these relationships. The authors must perform a sensitivity analysis to test the model's robustness in scenarios where these historical correlations break down.

We agree that the effects of these regressions should be explored in the manuscript. We have now changed Figure S1 (new version below) to include the future across all 1703 scenarios in the IPCC AR6 database with the species for FaIR. The left column contains the data from the prior figure with these future extensions. In the right column, we apply the errors in these fits to SSP245 in FaIR - i.e. we add the error in emissions (or forcing for BC on Snow) to the default SSP245 median run in FaIR. We do this for the 5, 10, 50, 90, 95 percentiles of the error, to

explore the potential effect across scenarios. The errors are overall slight - under 0.07K in each case and normally much lower than this.

[Figure]

*Figure S1: Regression predictions of NOx, VOC, and CO emissions, and the BC Snow forcing, and the effect of using these fits on temperature projections; see Section 2.1 for details on the targets and choice of predictors. Left: historical values of both the target variable (blue) and the emulation estimate when building a regression based on the predictor(s) (orange) are shown to*

*2015; this historical dataset is used in the regression. These are extended to 2100 for the 1703 scenarios in the IPCC AR6 database (Byers et al., 2022) which include all species input to the FaIR simple climate model, with IAM-reported emissions and their approximation using the regression shown. The 5th, 10th, 50th, 90th, and 95th percentiles of this emulation error are then applied to the default single-run SSP245 scenario in FaIR, with the resultant impacts on surface temperature shown on the right, using the same scale for each. Errors are less than a few hundredths of a degree throughout the period.*

We have added to the paragraph on L177 in the main text where this Figure is referred to, which now reads:

*Using historical data, and checked for plausibility using the future scenarios database, several candidate regression models were considered for each variable, with a linear relationship selected. Figure S1 shows their performance when used to estimate emissions in 1703 future scenarios from the IPCC AR6 database, and the effect of these future errors on temperatures in a medium-emissions scenario; errors are typically under a few hundredths of a degree.*

2- Unexplained Temperature Divergence from FaIR In high-emission scenarios (SSP5-8.5), FRIDA-Clim projects long-term temperatures roughly 0.5 K lower than the FaIR model. The authors attribute this to a "more responsive" process-based carbon cycle, but lack a mechanistic explanation. The authors should isolate whether this cooling is driven by the carbon cycle or the EBM parameterization to validate the model's physical plausibility.

We agree that exploring this effect in more detail is of important use here. To this end, we have now added a new Figure S6 exploring some new analysis to explore the relative contribution of carbon cycle and EBM contributions to these differences:

[Figure]

GMST response in FRIDA-Clim and FaIR under varying CO2 concentration in same EBM (top), identical forcings in separate EBMs (bottom)

*Figure S6: investigation of the relative contribution of differences in carbon cycle and EBM parameters to the difference between FRIDA-Clim and FaIR in the SSPs. Top row: GMST when passing the CO2 concentrations in the tow row of Figure S5 to the same (default FaIR) EBM with all other species as before; differences here are therefore due to the effect of CO2 concentration variation. Bottom row: GMST when running the same forcing (from RFMIP) in each posterior ensemble (i.e. just the EBM); differences here are due to the EBM variations. Overall, comparing to Figure 6, it appears the stronger carbon sinks in FRIDA-Clim drive the cooler response in overshoot scenarios, while a more sensitive EBM causes FaIR's higher temperatures in high emissions scenarios.*

We have now added to the main text on L806 to describe this result:

*In high emissions scenarios, FaIR's temperature stabilises at consistently greater levels. Further analysis assessing the contribution from differing carbon cycle and EBM responses to the GMST differences (Figure S6) suggests that FRIDA-Clim exhibits stronger carbon sinks and a less sensitive EBM than FaIR, with the former effect driving the inter-model variation in low-emissions scenarios and the latter effect responsible for the high-emissions differences.*

Overall, we find that both the carbon cycle and EBMs feature differences, with the former causing a larger effect on low-emissions scenario and the latter on high-emissions. Since they are calibrated on the same GMST data this suggests an interesting trade-off caused by the different model structures here. We feel that this is very useful context to the readers, but that further investigation into these differences may detract from the broader purpose of this manuscript, and is best left to the next round of RCMIP, for which the protocol is now available

and will include a significant focus on the differences between carbon cycle models in different ESMs (Romero-Prieto et al., 2025) and to which we have therefore referred now in this paper.

3- In Section 2.4.3, the definition of "FLU emissions" includes the realized soil carbon changes from land use transitions. Please clarify if this accounts for the committed emissions instantly or if it tracks the legacy flux over time. The distinction is important for annual budget accounting in the IAM.

We agree that more information is useful here. We have added the following explanation to the text (L471):

*As separate human reservoirs are not included for every soil carbon pool – which would allow the individual tracking of FLU emissions from land use transitions – the annual realized emissions induced by the transitions are approximated in a two-stage process. Firstly, the committed CO2 emissions from land use transitions are calculated using the difference in average soil carbon per area between the two land use types. These committed CO2 emissions from each transition type add to a stock of global committed future soil carbon loss from land use transitions. Secondly, this global stock of committed soil CO2 emissions decays with a rate of 0.1 yr-1. The annual reduction of this stock through the decay is then taken to be the realized annual soil carbon change induced by land use transitions. This implies that committed emissions from land use transitions are tracked via an approximated legacy flux. It should be noted that the separation of the TCB into a natural and a human component represents a pure accounting exercise, and that the actual annual change in terrestrial carbon, the TCB, is determined by the vegetation and soil carbon processes described in the sections above.*

Minor note: The use of the "IAM" abbreviation in the title may be obscure to readers outside the immediate integrated assessment community. I suggest spelling it out to improve accessibility and discoverability.

Reviewer 1 also made this comment and we agree entirely, and have made this change.

*Romero-Prieto, A., Sandstad, M., Sanderson, B. M., Nicholls, Z. R. J., Steinert, N. J., Gasser, T., Mathison, C., Kikstra, J., Aubry, T. J., and Smith, C.: Reduced Complexity Model Intercomparison Project Phase 3: experimental protocol for coordinated constraining and evaluation of reduced-complexity models, EGUsphere [preprint], https://doi.org/10.5194/egusphere-2025-5775, 2025.*